# Comparison of Operating Conditions, Postoperative Pain and Recovery, and Overall Satisfaction of Surgeons with Deep vs. No Neuromuscular Blockade for Spinal Surgery under General Anesthesia: A Prospective Randomized Controlled Trial

**DOI:** 10.3390/jcm8040498

**Published:** 2019-04-12

**Authors:** Seok Kyeong Oh, Woo-Keun Kwon, Sangwoo Park, Sul Gi Ji, Joo Han Kim, Youn-Kwan Park, Shin Young Lee, Byung Gun Lim

**Affiliations:** 1Department of Anesthesiology and Pain Medicine, Korea University Guro Hospital, Korea University College of Medicine, Seoul 08308, Korea; nanprayboy@korea.ac.kr (S.K.O.); parksangwoo1013@naver.com (S.P.); rita0511@naver.com (S.G.J.); 2Department of Neurosurgery, Korea University Guro Hospital, Korea University College of Medicine, Seoul 08308, Korea; kwontym@gmail.com (W.-K.K.); nskjh94@korea.ac.kr (J.H.K.); ykapa76@yahoo.co.kr (Y.-K.P.); 3Department of Anesthesiology and Pain Medicine, Bundang Jesaeng Hospital, Seongnam, Gyeonggi-do 13590, Korea; prologue02@gmail.com

**Keywords:** neuromuscular blockade, neuromuscular monitoring, neurosurgical procedures

## Abstract

We aimed to investigate operating conditions, postoperative pain, and overall satisfaction of surgeons using deep neuromuscular blockade (NMB) vs. no NMB in patients undergoing lumbar spinal surgery under general anesthesia. Eighty-three patients undergoing lumbar fusion were randomly assigned to receive deep NMB (*n* = 43) or no NMB (*n* = 40). In the deep-NMB group, rocuronium was administered to maintain deep NMB (train-of-four count 0, post-tetanic count 1–2) until the end of surgery. In the no-NMB group, sugammadex 4 mg/kg at train-of-four (TOF) count 0–1 or sugammadex 2 mg/kg at TOF count ≥2 was administered to reverse the NMB 10 min after placing the patient prone. Peak inspiratory airway pressure, plateau airway pressure, lumbar retractor pressure significantly were lower in the deep-NMB group. Degree of surgical field bleeding (0–5), muscle tone (1–3), and satisfaction (1–10) rated by the surgeon were all superior in the deep-NMB group. Pain scores, rescue fentanyl consumption in post-anesthesia care unit (PACU), and postoperative patient-controlled analgesia consumption were significantly lower in the deep-NMB group, and this group had a shorter length of stay in PACU. Compared to no NMB, deep NMB provides better operating conditions, reduced postoperative pain and higher overall satisfaction in lumbar spinal surgery.

## 1. Introduction

Since the introduction of sugammadex, which provides more predictable and rapid recovery from deep neuromuscular blockade (NMB) [1,2], questions have emerged regarding the issue of maintaining deep NMB until the very end of surgery [3,4,5]. Previous studies regarding the potential advantages of deep NMB were mainly performed in laparoscopic abdominal/pelvic surgeries [3,5,6,7,8] that require full relaxation of the diaphragm and abdominal muscles, which are very resistant to neuromuscular blocking agents [9,10]. In contrast, spinal surgery is often performed using no, shallow or moderate NMB in current clinical practice; thus, few reports have investigated the effect of deep NMB on operating conditions.

Absent or insufficient NMB may cause detrimental effects during spine surgery, including deterioration of the surgical environment for microscopic procedures due to increased muscle tone and inadvertent patient movement. In addition, this can cause an increase in bleeding due to increased intrathoracic and abdominal pressure in a prone position which forces blood from the inferior vena cava into the extradural venous plexus [11]. Also, the prone position for spinal surgery during general anesthesia may be associated with increased airway pressure. Increases in peak inspiratory airway pressure related to high muscle tone can lead to increased surgical bleeding [12,13], which can worsen operating condition.

There is no standardized method of anesthesia for spine surgery. Various anesthetic methods and the variable intensity of NMB through diverse strategies provide various operating conditions. Nevertheless, few studies have compared the operating conditions for spine surgery based on the degree of NMB. Therefore, we aimed to investigate not only the variables affecting operating conditions including peak inspiratory airway pressure, back muscle tone, bleeding in the surgical field and overall satisfaction of surgeons, but also postoperative pain and recovery between deep NMB and no NMB for spinal surgery under general anesthesia. We hypothesized that deep NMB could provide better surgical conditions, higher surgeon satisfaction, comparable postoperative recovery profiles, and lower postoperative pain compared to no NMB.

## 2. Materials and Methods

### 2.1. Patients and Study Design

This study was approved by the Korea University Guro Hospital Institutional Review Board, Seoul, Republic of Korea (KUGH15297-002), and registered at ClinicalTrials.gov (NCT02724111), and followed Consolidated Standards of Reporting Trials (CONSORT) 2010 guidelines (Appendix A). This study was performed at Korea University Guro Hospital, Seoul, South Korea, from May 2016 to January 2017 as a single-center prospective double-blinded randomized controlled trial. Written informed consent was obtained from every patient who was scheduled to undergo elective lumbar fusion surgery, was aged 18–75, and had an American Society of Anesthesiologists (ASA) physical status I–II. Patients were excluded if they had a known hypersensitivity to the drugs used in this study, known neuromuscular disease, significant liver or renal dysfunction, or a body mass index >30.0 kg/m^2^. Patients with hemodynamic instability (change in mean blood pressure >30% from baseline for >15 min or blood loss >1 liter during surgery) were also excluded.

Participants were randomly assigned to either deep NMB or no NMB group by a web-based computer-generated list and were unaware of their assignment. The randomized-numbers were kept in opaque sealed envelopes that were opened in the operating room by independent anesthesiologists not involved in the study.

The investigators who assessed the study endpoints were blinded to the group assignment. Only independent anesthesiologists who performed the induction and maintenance of anesthesia and NMB management were not blinded.

### 2.2. Anesthetic Regimen and NMB Management

All patients were premedicated with intramuscular midazolam 2 mg 30 min before anesthesia. Bispectral index (BIS, BIS VISTA^™^; Aspect Medical Systems Inc., Norwood, MA, USA) was established to monitor the depth of hypnosis. Neuromuscular function was monitored by train-of-four (TOF) stimulation and post-tetanic count (PTC) using a TOF-Watch SX (Organon Ireland Ltd, Schering-Plough Corporation, Dublin, Ireland). The forearm was immobilized with a detachable wrist brace (Neoban wrist support, Seoul Brace, Seoul, Korea) that allowed the thumb to move, with the stimulating electrode attached to the passage of the ulnar nerve and the elastic preload positioned at the thumb to monitor the response of the adductor pollicis (Appendix B
Figure A1). In the prone position, both arms and hands of all patients were placed forward on the armrests, which made sure the thumb of the patient could be freely moved and was not restricted by anything, including the surgical drapes. A large screen was established for masking groups from the surgeons just after placing the patient prone (Appendix B
Figure A2). Neuromuscular management and monitoring were performed according to the Good Clinical Research Practice guidelines [14].

Anesthesia was induced and maintained with total intravenous anesthesia using target-controlled infusion of propofol and remifentanil. In all patients, an effect-site concentration of propofol was increased stepwise from the initial 3.0 μg/mL until arrival of BIS <60, and then the TOF-Watch SX was stabilized and calibrated using a 5-s, 50-Hz tetanic stimulus and the automatic calibrating procedure [4]. Then, remifentanil infusion was started at an effect-site concentration of 1.5 ng/mL and rocuronium 1 mg/kg was injected and intubation was performed at TOF count 0. TOF and PTC stimulation were repeated every 15 s and 3–4 min, respectively.

To maintain the BIS value 40–60 as possible, the effect-site concentration of propofol was titrated upward or downward by 0.5–1.0 μg/mL. If hypertension/tachycardia or hypotension/bradycardia persisted for more than 5 minutes, the effect-site concentration of remifentanil was adjusted with an increase or decrease of 0.5–1.0 ng/mL. Despite these adjustments, if mean arterial pressure decreased or increased by more than 30% from the baseline, ephedrine 4 mg or nicardipine 0.5 mg, respectively, was injected. Through an esophageal thermometer, the core temperature was maintained around 36 °C using a warm air blower.

The ventilator was initially set at a tidal volume of 8 mL/kg and a respiratory rate of 12 breaths/min with oxygen in air (FIO_2_ = 0.5) flow of 3 L/min under volume-controlled ventilation, then the respiratory rate was controlled using an end-tidal carbon dioxide pressure ranging 30–35 mmHg through capnography. A positive end-expiratory pressure of 4 cmH_2_O, a 10% inspiratory pause time, and an inspiratory-to-expiratory ratio of 1:2 were applied.

The treatment of NMB was as follows. In the deep NMB group, rocuronium was administered with an infusion pump to maintain deep NMB (TOF count 0, PTC 1–2) until the end of surgery. The rocuronium infusion was composed of 2 mg/mL rocuronium diluted in 0.9% isotonic saline and started at 15 mL/h then titrated with close observation, and a saline bolus was injected 10 min after placing the patient prone to mask the group assignment from surgeons. In the no-NMB group, 0.9% isotonic saline was infused until the end of surgery and titrated randomly at the discretion of the attending anesthesiologist, and sugammadex 4 mg/kg at TOF count 0–1 or sugammadex 2 mg/kg at TOF count 2 or more was planned to be administered to reverse the NMB 10 min after placing the patient prone. After this, rocuronium was not used during the surgery unless the patient showed body movement during surgery or surgeons expressed complaints about high muscle tone; under these conditions, rescue rocuronium 5 mg was administered and body movement and the dose of administered rocuronium were recorded.

### 2.3. Surgical Procedure

Surgical lumbar retractor blades (Karlin^™^ Crank Frame Spinal Retractor Set, Codman, Germany) were positioned at the middle of the incision, lying just above the target level facet joint. All surgical procedures were performed by two experienced spine surgeons to minimize possible heterogeneity in opening techniques while avoiding the risk of bias caused by a single investigator.

A flat planar pneumatic pressure transducer (Catheter ICP monitor probe, Spiegelberg GmbH, Hamburg, Germany) was placed beneath the right-sided lumbar retractor blade; for two-level surgeries, the transducer was placed beneath the left cranial side blade in all cases. The distal tip of the transducer was positioned at the middle of the muscle retractor to achieve maximal contact with the multifidus muscle (Figure 1), and then the lumbar retractor blades were retracted until they reached the lateral margin of the bilateral facet joints, providing a full view of the operative lumbar level (Appendix B
Figure A3).

### 2.4. Reversal of NMB, Postoperative Recovery and Pain Management

At the end of the surgical procedure, fentanyl 50 µg and ramosetron 0.3 mg were injected. The surgeon who assessed the operating conditions left the operating room at the end of surgery, and the non-blinded anesthesiologist assessed emergence profiles. After returning the patient supine, propofol/remifentanil was discontinued and sugammadex 4 mg/kg was injected to reverse NMB of the patients in the deep-NMB group. In the no-NMB group, no reversal agent was injected if the TOF ratio was ≥0.9; otherwise, pyridostigmine 10 mg and glycopyrrolate 0.4 mg were injected. Emergence profiles were measured as elapsed times after discontinuation of anesthetics.

After arriving at the post-anesthesia care unit (PACU), the pain score was evaluated according to a numerical rating scale (NRS; 1–10) every 10 min, and fentanyl 10 µg was administered if the NRS was >3. The sedation score was assessed by the Modified Observe Assessment Alertness and Sedation Score (MOAA/SS) [15]. The patients were transferred to the ward when the NRS was <4 and the MOAA/SS was 5. Intravenous patient-controlled analgesia (PCA; fentanyl 600 µg, hydromorphone 6 mg and nefopam 100 mg, total volume 60 mL with normal saline; 0.5 mL/h basal rate, 0.5 mL bolus dose, 15 min lockout period) was started after the patient arrived at the ward to avoid the risk of opioid-induced respiratory complication during the patient’s transport.

### 2.5. Evaluation of Operating Conditions and Other Intraoperative Outcomes

Outcomes measured included: (1) the mean value of peak inspiratory airway pressure (primary endpoint) and plateau airway pressure measured every 15 min during surgery; (2) the muscle tone scaled by surgeons (1–3: 1 = good, suitable for surgery; 2 = moderate, but did not affect the operation; and 3 = hard, making the operation difficult) [16] during placement of the back muscle retractor, screw insertion through the pedicle of spine, and the overall period; (3) the mean value of the back muscle retractor pressure every 15 minutes; (4) the number of body movements per patient; (5) the degree of bleeding in the surgical field on microscopic view by the surgeons (no bleeding (0) to severe bleeding (5)) [17]; (6) overall satisfaction scaled by the surgeons (1–10; no satisfaction (1) to absolute satisfaction (10)); (7) cumulative dose of rocuronium (induction, rescue, and total) and average infusion rate of anesthetics; and (8) TOF count/ratio and BIS scores, mean arterial pressure and heart rate at the main time points (baseline, after intubation, skin incision, 30, 60 and 120 min after skin incision and the end of surgery).

Two blinded surgeons assessed the operating conditions including the muscle tone, degree of bleeding, overall satisfaction, and the patient’s body movement.

### 2.6. Evaluation of Postoperative Outcomes

The postoperative outcomes included: (1) emergence profiles including the time to recovery of spontaneous respiration, eye opening, and extubation; (2) recovery time in the PACU (the time to the MOAA/SS of 5 from entering the PACU); (3) length of stay and (4) rescue fentanyl consumption in the PACU; (5) cumulative consumption of PCA on the ward; (6) postoperative pain scores (NRS; 1–10); and (7) the occurrence of adverse events (e.g., respiratory depression, oversedation, hypersensitivity, tachycardia, bradycardia, hyper- or hypotension, headache, dizziness, and difficulty urinating) in the PACU and on the ward over the first 48 h after surgery.

### 2.7. Statistical Analysis

The primary endpoint in this study was the mean value of peak inspiratory airway pressure measured every 15 min during surgery. The sample size calculation was based on a pilot study with 10 cases in each group. In the pilot study, the peak inspiratory airway pressure was 18.8 ± 2.9 cmH_2_O in the deep-NMB group and 20.7 ± 3.0 cmH_2_O in the no-NMB group. The effect size was 0.64. On the assumption that the allocation ratio of the two groups was 1, a sample size of 38 patients was selected for each group, calculated by a two-sided Student’s *t*-test with a significance level of 0.05 and a power of 0.8. We estimated a 15% dropout rate, resulting in the final enrollment of 45 patients in each group.

Statistical analyses were performed using the SPSS software (version 20.0; IBM, Armonk, NY, USA). The normal distribution of continuous data was first evaluated using the Shapiro–Wilk test (*p* > 0.05). The normally distributed data were analyzed using Student’s *t*-test, and the abnormally distributed data were analyzed using Mann–Whitney *U* test.

The Student’s *t*-test or Mann–Whitney *U* test were used to compare the age, height, weight, anesthesia time, operation time, mean values of peak inspiratory airway pressure, plateau airway pressure, and retractor pressure, emergence times, cumulative dose of rocuronium or average infusion rate of anesthetics, rescue fentanyl consumption, recovery time and length of stay in the PACU, and cumulative PCA consumption.

Ordinal parameters including the bleeding scale and overall satisfaction score scaled by the surgeon were compared using the Mann–Whitney *U*-test, while categorical variables including sex, ASA grade, operation level, muscle tone, number of body movements per patient, and the incidence of adverse events were compared by a chi-square test or Fisher exact test. The changes over time in the TOF ratio and BIS values, mean arterial pressure and heart rate during surgery, the NRS for pain in the PACU and on the ward were compared using repeated measures analysis of variance.

Data are expressed as mean ± standard deviation, median (range), or number of patients (%). *P*-values were two-tailed, and a *p*-value < 0.05 was considered statistically significant.

## 3. Results

A total of 90 patients were enrolled, and 7 patients were excluded due to the discontinued intervention during operation. Finally, 83 patients were evaluated (Figure 2). There was no significant difference between the two groups in baseline patient characteristics, operation time, anesthesia time, and the number of operated spinal levels (Table 1).

Peak inspiratory airway pressure, plateau airway pressure, and retractor pressure were significantly lower in the deep-NMB group than in the no-NMB group (18.4 ± 1.1 vs. 20.2 ± 1.1, 17.1 ± 1.4 vs. 19.4 ± 1.1 cmH_2_O and 81.2 ± 9.1 vs. 100.0 ± 7.3 mmHg, respectively) (*p* < 0.001). The operating conditions and overall satisfaction score (8.0 ± 1.3 vs. 3.1 ± 1.2; *p* < 0.001) evaluated by surgeons were superior in the deep-NMB group (Table 2).

Rescue rocuronium consumption was significantly higher in the no-NMB group than deep NMB (15.1 ± 9.4 vs. 0.6 ± 1.6 mg; *p* < 0.001) (Table 2). Throughout the surgery, five patients in the deep-NMB group received a single injection of rescue rocuronium (5 mg), while in the no-NMB group, 37 patients received 1–7 rescue rocuronium injections (5–35 mg). All patients in the no-NMB group received sugammadex 4 mg/kg 10 min after placing the patient prone because all patients had a TOF count of 0 at that time (the time from rocuronium injection at induction to sugammadex injection: 22 ± 4 min), and 3 patients in the no-NMB group received pyridostigmine 10 mg and glycopyrrolate 0.4 mg for reversal due to a TOF ratio less than 0.9 at the end of surgery. The average infusion rate of propofol was significantly lower in the deep-NMB group (0.104 ± 0.014 vs. 0.113 ± 0.015 mg kg^−1^ min^−1^; *p* < 0.001), but there was no difference in the average infusion rate of remifentanil (Table 2).

The change over time in the TOF ratio was significantly different between the groups (Figure 3a; group: *F*(1,81) = 11,692, *p* < 0.001, time: *F*(2.7,486) = 72,664, *p* < 0.001, group*time: *F*(2.7,486) = 4622, *p* < 0.001), but changes over time in the BIS, mean arterial pressure, and heart rate were not different between the groups (Figure 3b; group: *F*(1,81) = 4.524, *p* = 0.36, time: *F*(4.2,486) = 629.174, *p* < 0.001, group*time: *F*(4.2,486) = 1.274, *p* = 0.279, Figure 3c; group: *F*(1,81) = 8.646, *p* = 0.004, time: *F*(3.6,486) = 115.116, *p* < 0.001, group*time: *F*(3.6,486) = 0.814, *p* = 0.507, Figure 3d; group: *F*(1,81) = 3.384, *p* = 0.69, time; *F*(3,486) = 82.345, *p* < 0.001, group*time; *F*(3,486) = 0.588, *p* = 0.624, respectively).

Recovery profiles during emergence from anesthesia in the operating room and recovery time in the PACU were not different (Table 3).

The NRS for pain in the PACU and rescue fentanyl consumption in the PACU were significantly lower in the deep-NMB group (Table 3; group: *F*(1,81) = 75.147, *p* < 0.001, time; *F*(1.6,162) = 464.766, *p* < 0.001, group*time; *F*(1.6,162) = 5.445, *p* = 0.010). The length of stay in the PACU was significantly shorter (61.9 ± 5.9 vs. 87.0 ± 24.5; *p* < 0.001) and the incidence of adverse hemodynamic events in the PACU was significantly lower in the deep-NMB group (Table 3). The change over time in the NRS for pain on the ward did not differ between the two groups (Table 3; group: *F*(1,81) = 2.652, *p* = 0.107, time: *F*(1.4,162) = 205.046, *p* < 0.001, group*time: *F*(1.4,162) = 2.736, *p* = 0.087). Cumulative PCA consumption during the postoperative 48 hours was significantly lower in the deep-NMB group (39.9 ± 11.9 vs. 51.2 ± 9.3; *p* < 0.001) (Table 3). The incidence of adverse events on the ward was comparable between the groups.

## 4. Discussion

In the present study, deep NMB offered many benefits compared to no NMB in spine surgery. A benefit of deep NMB is the association with lower airway pressure. In this study, deep NMB decreased the airway pressure including peak inspiratory airway pressure and plateau airway pressure compared to no NMB during mechanical ventilation. During spine surgery in the prone position under general anesthesia, airway pressure usually increases due to the increased tone of the abdominal and respiratory muscles, including the diaphragm, resulting in increased intrathoracic and intra-abdominal pressure. These consequences may bring about similar detrimental results to those derived from an increase in intra-abdominal pressure in laparoscopic abdominal surgeries, including the potential for barotrauma, decrease in tidal volume/minute ventilation, and a potential increase in intracranial pressure and intraocular pressure [18,19]. In addition, it may worsen the surgical view since the pooling in Batson’s plexus around the spine can increase bleeding ring surgery [11]. Previous studies reported that decreased intraoperative surgical bleeding may be related to lower intraoperative peak inspiratory airway pressure in lumbar fusion surgery [12,13]. Reduced bleeding can facilitate adequate surgical field visualization during operation, which is especially important when the surgical procedure is performed under microscope. Therefore, airway pressure can be significantly associated with the operating condition, especially as the degree of surgical field bleeding as evaluated by the surgeon was significantly lower in the deep-NMB group.

Surgeons and patients were more subjectively and objectively satisfied with deep NMB than with no NMB. The surgeons could perform operations in comfort due to lower muscle tone (subjective) and lower retraction pressure (objective) in deep NMB. Similarly, patients complained of less pain (subjective), consumed fewer postoperative opioids (objective), and needed a shorter length of stay in the PACU (objective). The longer length of stay in the PACU in the no-NMB group may have been due to the need for more rescue fentanyl administration and a higher infusion rate of propofol during surgery. Despite adjustment of the BIS to the range of 40–60 during surgery, a higher average infusion rate of propofol was observed in the no-NMB group than in the deep-NMB group. The greater propofol administration by anesthesiologists observed in the no-NMB group might have been done to avoid complications or surgical procedure disturbance due to body movement during surgery. Nevertheless, the difference was not considered clinically significant because the BIS value during surgery and the recovery profiles during emergence from anesthesia were comparable between the two NMB groups. In addition, adverse hemodynamic events in the PACU were higher in the no-NMB group, which could be explained by higher pain and recue opioid consumption in the PACU.

These results are identical to those of recent studies, which presented potential advantages of deep NMB in laparoscopic surgery [3,5,6,7,8,20]. During pneumoperitoneum in laparoscopic surgeries, deep NMB could provide better visibility and maintain lower carbon dioxide insufflation pressure, which reduces the incidence of postoperative pain [5,20,21,22].

On the contrary, few reports describe the impact of muscle relaxation in non-laparoscopic surgery [4,16,23,24,25,26,27]. Moreover, to our knowledge, there is just one report that explores the effectiveness of deep NMB in spinal surgery. Li et al. [16] suggested that total intravenous anesthesia without a muscle relaxant provided similar operating conditions to those observed with a muscle relaxant in spine surgery with regard to muscle tone, airway pressure and body movements, which implies that NMB may be unnecessary in spinal surgery. However, in their study, the group with muscle relaxants did not receive standardized deep NMB. The TOF count was 0, but PTC was not evaluated, and details regarding the total administered dose or frequency of muscle relaxants were not described. Therefore, as the study did not follow the criteria for deep NMB (TOF count 0, PTC 1–2) applied to the current controlled NMB trials [3,4,5,7,28], we established the standardized deep NMB group based on these criteria.

An important strength of this study was the measurement of retraction pressure against the paraspinal muscles using a pneumatic pressure transducer, which attempted to objectively assess surgical conditions. To our knowledge, no previous studies have investigated the retraction pressure to measure the depth of NMB in spine surgery, though there have been numerous studies about the correlation between pneumoperitoneum pressure and the depth of NMB in laparoscopy [7,8,28,29]. We have provided the first evidence for the usefulness of deep NMB in spine surgery in an objective manner using retractor pressure. In the present study, deep NMB showed a statistically significant decrease in intraoperative retraction pressure and led to lower postoperative back pain. These positive results in deep NMB could be helpful for researchers to plan relevant future studies.

In classical posterior lumbar operations, dissection and retraction of the paraspinal muscles are inevitable and always result in some degree of muscle injury. Several previous studies have revealed the histological, histochemical and enzymatic adverse effects of lumbar muscle retraction after back surgery [30,31,32,33]. Muscle dissection itself, accompanied by additional retraction injuries, can lead to denervation of the muscles [34], as well as increased muscle pressure as a consequence of alterations in muscle blood flow [33,35]. Kawaguchi et al. [31] reported that back muscle injuries were related to the retraction pressure and its time and extent of exposure, and marked edema and fiber necrosis were obvious and creatine phosphokinase activity tended to be high as the pressure/time of retraction increased. These previously documented negative effects of lumbar retraction pressure on the paraspinal muscles might be one origin of post-operative pain after spinal surgery. This could be diminished by deep NMB, as it reduced retraction pressure in our study.

This study has some limitations. First, the potential for worsened surgical conditions due to higher muscle tone in the no-NMB group could have been predicted, and the investigation on them in moderate NMB rather than no NMB seems to be of more interest in terms of clinical application; nonetheless, no NMB in the present study reflects the NMB state in the common clinical practice in which total intravenous anesthesia with motor-evoked potential monitoring is used for spine surgery [36,37]. In addition, there have been no prior comparisons of deep vs. no NMB in spinal surgery; therefore, as the first study, we sought to determine whether deep NMB in spinal surgery was effective. Second, there was a possibility that due to the learning curve, surgeons could gradually recognize and predict the degree of NMB using their experience to classify the degree of muscle twitch during electrical cautery. Therefore, we tried to prevent the potential learning effect and maintain consistent assessments of operating conditions using a prearranged plan in which two blinded surgeons rather than a single surgeon assessed the operating conditions. Third, the advantages of deep NMB could be compromised when intraoperative neuromonitoring is necessary in spine surgery, because the neuromuscular blocking agents should be minimized or avoided during the neuromonitoring, such as motor evoked potential. Intraoperative neuromonitoring is helpful in revealing structural compromise along the sensory and motor pathways and is an integral component to a variety of spinal procedures. Nevertheless, neuromonitoring is not always indicated in all forms of spine surgery, such as lumbar fusion of 1 or 2 levels, which was included in our study. Therefore, the advantages of deep NMB in spinal surgery are significant in many cases without neuromonitoring.

## 5. Conclusions

Deep NMB provided better physiologic and operating conditions during surgery and reduced postoperative pain, analgesic consumption, length of stay in the PACU, and PCA consumption during the first 48 h after surgery compared to no NMB in patients undergoing lumbar fusion surgery under general anesthesia.

## Figures and Tables

**Figure 1 jcm-08-00498-f001:**
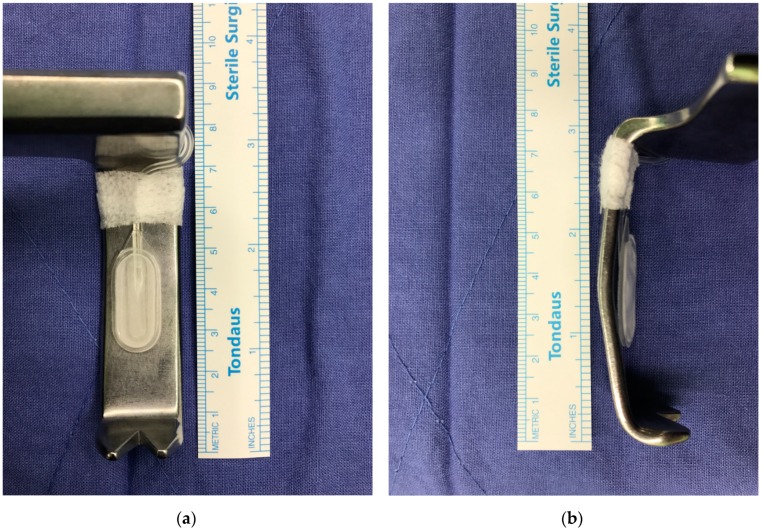
The pressure transducer used for retractor pressure monitoring was attached to a retractor blade. It was mounted on the middle of the inner surface of the retractor. (**a**) Medial view. (**b**) Lateral view.

**Figure 2 jcm-08-00498-f002:**
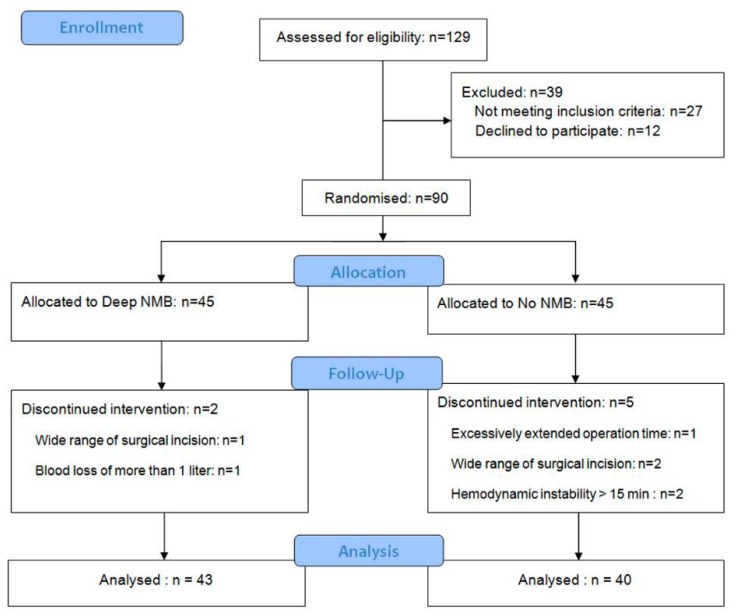
A flow-chart describing patient recruitment, randomization, and withdrawal. NMB: neuromuscular blockade.

**Figure 3 jcm-08-00498-f003:**
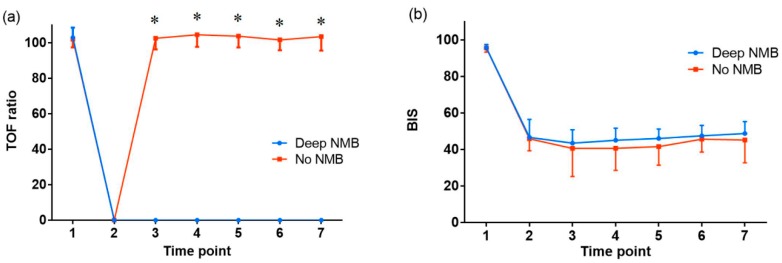
The change over time in (**a**) train-of-four (TOF) ratio, (**b**) bispectral index (BIS), (**c**) mean arterial pressure and (**d**) heart rate at the main time points during the perioperative period. NMB: neuromuscular blockade. The graphs show the mean value and standard deviation of each variable for each time point during general anesthesia. Time point 1: at baseline (before anesthesia), 2: at intubation, 3: at skin incision, 4: 30 min after skin incision, 5: 60 min after skin incision, 6: 120 min after skin incision, 7: at the end of surgery. * *p* < 0.001.

**Table 1 jcm-08-00498-t001:** Patient characteristics and clinical data.

	Deep NMB	No NMB
	(*n* = 43)	(*n* = 40)
Age (year)	61 (37–75)	60 (30–75)
Sex (male/female)	12/31	11/29
ASA (I/II)	11/32	7/33
Height (m)	1.59 ± 0.07	1.58 ± 0.08
Weight (kg)	59.8 ± 8.5	60.0 ± 6.9
Operation level (1/2)	36/7	36/4
Operation time (min)	221.9 ± 48.6	220.7 ± 51.3
Anesthesia time (min)	263.6 ± 53.6	257.6 ± 53.2

Values are mean ± standard deviation, median (range) or number of patients; NMB, neuromuscular blockade; ASA, American Society of Anesthesiologists physical status classification.

**Table 2 jcm-08-00498-t002:** Intraoperative outcomes including airway pressures and variables measured for operating condition assessment, and cumulative dose of rocuronium and average infusion rate of anesthetics.

	Deep NMB	No NMB	*p*-Value
	(*n* = 43)	(*n* = 40)	
Variables for operating condition			
Peak inspiratory airway pressure (cmH_2_O)	18.4 ± 1.1	20.2 ± 1.1	<0.001
Plateau airway pressure (cmH_2_O)	17.1 ± 1.4	19.4 ± 1.1	<0.001
Retractor pressure (mmHg)	81.2 ± 9.1	100.0 ± 7.3	<0.001
Surgical field bleeding scale (0/1/2/3/4/5)	0/14/23/5/1/0	0/1/4/19/16/0	<0.001
Retractor placement muscle tone (1/2/3)	29/8/6	0/4/36	<0.001
Screw insertion muscle tone (1/2/3)	37/1/5	5/10/25	<0.001
Other muscle tone (1/2/3)	31/12/0	4/26/10	<0.001
Body movements (number per patient)	0 (0–0)	2 (0–4)	<0.001
Surgical satisfaction (1–10)	8.0 ± 1.3	3.1 ± 1.2	<0.001
Administered dose of drugs			
Total rocuronium (mg)	232.4 ± 71.5	75.4 ± 13.2	<0.001
Induction rocuronium (mg)	59.9 ± 8.3	60.0 ± 6.9	0.571
Rescue rocuronium (mg)	0 (0–5)	15 (0–35)	<0.001
Remifentanil average rate (μg kg^−1^ min^−1^)	0.035 ± 0.017	0.041 ± 0.020	0.167
Propofol average rate (mg kg^−1^ min^−1^)	0.104 ± 0.014	0.113 ± 0.015	<0.001

Values are mean ± standard deviation, median (range) or number of patients; Surgical field bleeding scale: 0—No bleeding, 1—Slight bleeding—no suctioning of blood required, 2—Slight bleeding—occasional suctioning required, 3—Slight bleeding—frequent suctioning of blood was required but not threatened the operative field, 4—Moderate bleeding—frequent suctioning of blood was required which moderately threatened the operative field, 5—Severe bleeding—very frequent suctioning of blood was required which severely threatened the operative field; NMB, neuromuscular blockade.

**Table 3 jcm-08-00498-t003:** Postoperative outcomes in the operating room, post-anesthesia care unit (PACU) and ward.

	Deep NMB (*n* = 43)	No NMB (*n* = 40)	*p*-Value
Operating room (recovery profiles during emergence from anesthesia)			
Time to the recovery of spontaneous respiration (min)	5.4 ± 3.8	6.1 ± 4.3	0.623
Time to the recovery of eye opening (min)	8.3 ± 4.3	8.9 ± 4.3	0.509
Time to extubation (min)	10.1 ± 3.9	10.7 ± 4.5	0.795
PACU			
Numeric rating scale for pain (0–10)			0.010
when arriving at PACU	6.2 ± 2.0	8.4 ± 0.6	
30 min after arriving at PACU	3.9 ± 1.9	6.9 ± 0.9	
60 min after arriving at PACU	2.5 ± 1.4	5.1 ± 1.4	
Recovery time (time to sedation score 5) (min)	10.5 ± 7.9	17.0 ± 11.4	0.060
Length of stay (min)	61.9 ± 5.9	87.0 ± 24.5	<0.001
Rescue fentanyl consumption (μg)	31.4 ± 24.4	86.3 ± 29.9	<0.001
Adverse events (Y/N)	0/43	8/32	0.002
Ward			
Numeric rating scale for pain (0–10)			0.087
6 h after operation	3.1 ± 1.1	3.5 ± 0.8	
24 h after operation	2.3 ± 1.2	2.3 ± 0.8	
48 h after operation	1.3 ± 0.7	1.8 ± 0.9	
Total PCA consumption (mL)	39.9 ± 11.9	51.2 ± 9.3	<0.001
Postoperative nausea or vomiting (severe/moderate/mild/none)	1/9/1/32	4/4/0/32	0.202
Other adverse events (Y/N)	0/43	4/36	0.050

Values are mean ± standard deviation or number of patients; Adverse events at PACU in the no-NMB group: hypertension 4, hypotension 4. Other adverse events at Ward in the no-NMB group: dizziness 4; NMB: neuromuscular blockade; PACU, post-anesthesia care unit; PCA, patient-controlled analgesia.

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
