# Peer review of "Comparison of Operating Conditions, Postoperative Pain and Recovery, and Overall Satisfaction of Surgeons with Deep vs. No Neuromuscular Blockade for Spinal Surgery under General Anesthesia: A Prospective Randomized Controlled Trial"

_jcm, 2019, doi:10.3390/jcm8040498_

Round 1
Reviewer 1 Report
This report describes a randomized, prospective trial of patients undergoing spine surgery. One group underwent surgery with deep neuromuscular blockade (NMB) (N = 43), and one without (N = 40). Outcome measures included peak inspiratory airway pressure, plateau airway pressure, lumbar retractor pressure, intraoperative bleeding, surgeon satisfaction, postoperative pain levels, postoperative opioid use, and time spent in the PACU.
The patients in the deep NMB group had significantly lower peak inspiratory airway pressures, lower plateau airway pressures, lower lumbar retractor pressure values, less intraoperative bleeding, reduced muscle tone, higher surgeon satisfaction, lower pain scores, reduced postoperative opioid consumption, shorter length of stay in the PACU, and higher patient satisfaction with their procedure. Of note, the deep NMB group experienced significantly fewer adverse hematologic events in the PACU as well.
These data suggest that there are a number of advantages to performing lumbar spine surgery with deep NMB. Unfortunately, the use of neuromonitoring is precluded with this anesthetic regimen, which can be a significant disadvantage depending on the type of spine surgery being performed. Neuromonitoring can be helpful in revealing structural compromise along the sensory and motor pathways and is an integral component to a variety of spinal procedures, especially those involving instrumentation. It is unclear whether the advantages described in this study would outweigh the drawback of losing the ability to perform effective neuromonitoring in these patients.
Author Response
Point 1: This report describes a randomized, prospective trial of patients undergoing spine surgery. One group underwent surgery with deep neuromuscular blockade (NMB) (N = 43), and one without (N = 40). Outcome measures included peak inspiratory airway pressure, plateau airway pressure, lumbar retractor pressure, intraoperative bleeding, surgeon satisfaction, postoperative pain levels, postoperative opioid use, and time spent in the PACU.
The patients in the deep NMB group had significantly lower peak inspiratory airway pressures, lower plateau airway pressures, lower lumbar retractor pressure values, less intraoperative bleeding, reduced muscle tone, higher surgeon satisfaction, lower pain scores, reduced postoperative opioid consumption, shorter length of stay in the PACU, and higher patient satisfaction with their procedure. Of note, the deep NMB group experienced significantly fewer adverse hematologic events in the PACU as well.
These data suggest that there are a number of advantages to performing lumbar spine surgery with deep NMB. Unfortunately, the use of neuromonitoring is precluded with this anesthetic regimen, which can be a significant disadvantage depending on the type of spine surgery being performed. Neuromonitoring can be helpful in revealing structural compromise along the sensory and motor pathways and is an integral component to a variety of spinal procedures, especially those involving instrumentation. It is unclear whether the advantages described in this study would outweigh the drawback of losing the ability to perform effective neuromonitoring in these patients.
Response 1: We appreciate your good suggestion. As you pointed, the advantages of deep NMB could be compromised when the neuromonitoring is imperative in spine surgery. However, neuromonitoring is not always indicated in all forms of spine surgery such as lumbar fusion of 1 or 2 levels, which was included in our study. According to the retrospective study by James et al. in 2014 [Neurosurg Focus. 2014;37(5):E10.] the use of intraoperative neuromonitoring was 12% in spinal procedures, although the percentage may be increased these days. Therefore, these advantages of deep NMB in spinal surgery can have significance in many cases without neuromonitoring.
We added about this topic according to your suggestion in the limitation in the discussion section as follows in page 10, line 360-368 in the revised version.
“Third, the advantages of deep NMB could be compromised when intraoperative neuromonitoring is necessary in spine surgery, because the neuromuscular blocking agents should be minimized or avoided during the neuromonitoring such as motor evoked potential. Intraoperative neuromonitoring is helpful in revealing structural compromise along the sensory and motor pathways and is an integral component to a variety of spinal procedures. Nevertheless, neuromonitoring is not always indicated in all forms of spine surgery such as lumbar fusion of 1 or 2 levels which was included in our study; therefore, the advantages of deep NMB in spinal surgery can have significance in many cases without neuromonitoring.”
Reviewer 2 Report
While the manuscript is well written and the various measures are well performed, there are several important issues regarding the interpretation of these findings.
Comments
1. The authors analyzed repeated measures such as the TOF ratio, BIS values, MAP, HR, and NRS using two-way repeated measures ANOVA. Therefore, they should provide specific p-value and F-value for groups, time and interaction between groups and time respectively.
2. The effect-site concentration of remifentanil of 1.5μg/ml is TOO low to perform intubation. In addition, remifentanil infusion rate of 0.035-0.04μg/kg/min is TOO low to maintain spinal surgery. Please explain about this magic.
Author Response
Point 1: The authors analyzed repeated measures such as the TOF ratio, BIS values, MAP, HR, and NRS using two-way repeated measures ANOVA. Therefore, they should provide specific p-value and F-value for groups, time and interaction between groups and time respectively.
Response 1:
We appreciate your insightful suggestion.
Here we provided the specific values for each variable. Also, we added the specific value in the method section in the revised version page 7-8.
TOF ratio; group: F(1,81)=11692, P <0.001, time: F(2.7,486)=72664, P <0.001, group*time: F(2.7,486)=4622, P <0.001< span="">
BIS values; group: F(1,81)=4.524, P=0.36, time: F(4.2,486)=629.174, P <0.001, group*time: F(4.2,486)=1.274, P=0.279
MAP; group: F(1,81)=8.646, P=0.004, time: F(3.6,486)=115.116, P <0.001, group*time: F(3.6,486)=0.814, P=0.507
HR; group: F(1,81)=3.384, P=0.69, time; F(3,486)=82.345, P <0.001, group*time; F(3,486)=0.588, P=0.624
NRS for pain in the PACU; group: F(1,81)=75.147, P <0.001, time; F(1.6,162)=464.766, P <0.001, group*time; F(1.6,162)=5.445, P=0.010
NRS for pain in the Ward; group: F(1,81)=2.652, P=0.107, time: F(1.4,162)=205.046, P <0.001, group*time: F(1.4,162)=2.736, P=0.087
Point 2: The effect-site concentration of remifentanil of 1.5μg/ml is TOO low to perform intubation. In addition, remifentanil infusion rate of 0.035-0.04μg/kg/min is TOO low to maintain spinal surgery. Please explain about this magic.
Response 2: We appreciate your good question.
As described in the method section, stabilizing and calibrating times for TOF-Watch SX were additionally necessary during propofol infusion before rocuronium administration for intubation, which prolonged the induction time. During the longer induction period before intubation, the blood pressure tended to be decreased. Also, the median age of patients enrolled in this study was 60 and 61 years old (elderly) in each group because the population included in the study was the patients with degenerative lumbar spine diseases. Therefore, it was considered safest to administer remifentanil under titration at a smaller dose. In addition, the preparation for study set up (for establishing large screen and monitoring outcomes) made the period before surgical incision longer, which provided the potential to lower blood pressure in elderly population. For these reasons, relatively low concentration of remifentanil was infused to maintain the hemodynamic stability.
Consequently, as shown in the Figure 3 (c and d), the blood pressure and heart rate were kept stable after intubation and during surgery.
Reviewer 3 Report
Overall this is a very well prepared and written manuscript. The study design seems sound and the results well reported. I have only a number of minor points I believe the authors should address:
Table 2: you state the primary endpoint as the mean difference in PIP between groups, and you report this effect appropriately. You then report a number of other results using t-tests or Mann-Whitney-U tests, but you don’t report any kind of correction for multiple comparisons. You could either combine a number of these tests into an ANVOA or two, or perform a Holm-Bonferroni correction. The simplest solution would be the latter.
Page 5, line 208: Presumably it wasn’t the patient, as such, who was non-compliant (as they were anaesthetised) but the patient’s care which did not conform to protocol?
Figure 3: The figure markings and key are very small, so it is hard to make out which lines are which.
END
Author Response
Point 1: Table 2: you state the primary endpoint as the mean difference in PIP between groups, and you report this effect appropriately. You then report a number of other results using t-tests or Mann-Whitney-U tests, but you don’t report any kind of correction for multiple comparisons. You could either combine a number of these tests into an ANVOA or two, or perform a Holm-Bonferroni correction. The simplest solution would be the latter.
Response 1: We appreciate your good suggestion.
Statistical significance was re-determined by adjusted alpha (α / k [number of hypothesis test]) corrected values using Bonferroni correction as you recommended [ref. S. Lee et al. What is the proper way to apply the multiple comparison test? Korean J Anesthesiol. 2018 Oct; 71(5): 353–360.].
Therefore, the adjusted P-value is 0.05/7 = 0.0071
Because 7 variables (mean values of peak inspiratory airway pressure, plateau airway pressure, and retractor pressure, cumulative dose of rocuronium, induction rocuronium dose, and average infusion rates of remifentnil and propofol) were analyzed using t-tests or Mann-Whitney-U tests in Table 2.
With the adjusted P-value of 0.0071, there were no changes in outcomes with significance, as the all variables with significance had showed the P-value under 0.001.
This method can be also applicable to Table 3.
7 variables (3 variables in recovery profiles during emergence from anesthesia, rescue fentanyl consumption, recovery time and length of stay in the PACU, and cumulative PCA consumption) were analyzed using t-tests or Mann-Whitney-U tests in Table 3.
With the adjusted P-value of 0.0071, there were no changes in outcomes with significance, as the all variables with significance had showed the P-value under 0.001.
Point 2: Page 5, line 208: Presumably it wasn’t the patient, as such, who was non-compliant (as they were anaesthetised) but the patient’s care which did not conform to protocol?
Response 2: I appreciate your good comment.
Non-compliance with the study protocol means the 7 cases of discontinuation of intervention due to the hemodynamic instability (change in mean blood pressure >30% from baseline for >15 minutes or blood loss >1 liter during surgery) as described in the exclusion criteria (in page 2, line 73,74) or wide range of surgical incision/excessively extended operation time by surgical factors, which could be confounding factors with heterogeneous condition to study outcomes.
As you pointed, the expression of non-compliance can be confusing. Therefore, we change the expression as follows in page 5, line 208-209 in the revised version.
“……and 7 patients were excluded for noncompliance with the study protocol.”
“……and 7 patients were excluded due to the discontinued intervention during operation.”
Point 3: Figure 3: The figure markings and key are very small, so it is hard to make out which lines are which.
Response 3: We appreciate your good suggestion.
I revised the Figure 3 as you recommended in page 7-8.